# Knowledge and Attitudes of Healthcare Professionals Regarding Perinatal Influenza Vaccination during the COVID-19 Pandemic

**DOI:** 10.3390/vaccines11010168

**Published:** 2023-01-12

**Authors:** Chrysoula Taskou, Antigoni Sarantaki, Apostolos Beloukas, Vasiliki Ε. Georgakopoulou, Georgios Daskalakis, Petros Papalexis, Aikaterini Lykeridou

**Affiliations:** 1Midwifery Department, University of West Attica, 12243 Athens, Greece; 2Molecular Microbiology & Immunology Laboratory, Department of Biomedical Sciences, University of West Attica, 11521 Athens, Greece; 3National AIDS Reference Centre of Southern Greece, University of West Attica, 12243 Athens, Greece; 4Department of Infectious Diseases-COVID-19 Unit, Laiko General Hospital, 11527 Athens, Greece; 51st Department of Obstetrics and Gynecology, Alexandra Hospital, National and Kapodistrian University of Athens, 15772 Athens, Greece; 6Unit of Endocrinology, 1st Department of Internal Medicine, Laiko General Hospital, Medical School, National and Kapodistrian University of Athens, 15772 Athens, Greece

**Keywords:** healthcare professionals, knowledge, attitudes, influenza vaccination, pregnancy

## Abstract

Immunizations during pregnancy are an important aspect of perinatal care. Although the influenza vaccine during pregnancy is safe, vaccination rates are low. According to research data, one of the reasons for the low vaccination rates among pregnant women is that they do not receive a clear recommendation from healthcare providers. This study aims to record the knowledge and attitudes about influenza vaccination and investigate healthcare professionals’ recommendations during the perinatal period. A cross-sectional study was conducted with convenience sampling in Athens, Greece. Our purposive sample included 240 midwives, Ob/Gs, and pediatricians. Data were collected using an appropriate standardized questionnaire with information about demographics, attitudes towards influenza vaccination, and knowledge about the influenza virus and peripartum vaccination. Statistical analysis was conducted using IBM SPSS-Statistics version 26.0. This study identifies the reasons for the lack of vaccine uptake including a wide range of misconceptions or lack of knowledge about influenza infection, lack of convenient access to get vaccinated, etc. Misconceptions about influenza and influenza vaccines could be improved by better education of healthcare workers. Continuing professional education for health professionals is necessary to improve the level of knowledge, prevent negative beliefs, and promote preventive and therapeutic practices.

## 1. Introduction

In December 2019, a new disease known as coronavirus disease 2019 (COVID-19) emerged as a major world threat. The pandemic, caused by the severe acute respiratory syndrome coronavirus 2 (SARS-CoV-2), has exposed vulnerable populations to an unprecedented global health crisis. Initially, the ability of SARS-CoV-2 to spread in the population was considered to be similar to that of the influenza virus [1]. Of note, the influenza A virus (novel H1N1 subtype) was first identified in April 2009 with the WHO raising the influenza pandemic alert to its highest level in June 2009. Since then, seasonal influenza continues to be a major worldwide health hazard [2].

Seasonal influenza is an acute respiratory illness caused by a group of RNA viruses (A, B, and C) [3]. The most common symptoms are those of the upper respiratory tract, such as cough, sore throat, and runny nose, which are also associated with general symptoms such as fever, headaches, myalgia, and weakness. Influenza can also cause severe complications including viral pneumonia and death, which occurs most frequently in certain groups of patients with underlying chronic illnesses classified as high-risk [4]. The CDC and the World Health Organization have added pregnant women to the high-risk groups for severe influenza illness [5].

Influenza during the perinatal period may therefore be associated with adverse maternal and prenatal outcomes due to the physiological changes and immune adaptations that occur during pregnancy [6]. A systematic review and meta-analysis of observational studies showed that there was a higher risk for hospitalization in pregnant versus non-pregnant women infected with influenza [7]. Interestingly, the risk of hospitalization was higher in the third trimester of pregnancy [8]. Ethnic minority background, obesity, diabetes mellitus or chronic cardiac or pulmonary disease, advanced maternal age (≥35 years), living in increased socioeconomic deprivation and working in healthcare or other public-facing occupations are important risk factors for more severe illness and pregnancy complications [9,10].

In addition to the adverse maternal outcomes, many studies have shown that influenza may also lead to neonatal complications such as preterm delivery, low birth weight, and occasionally neonatal death [11]. In a recent Korean study, children born to women with influenza were at an increased risk of preterm birth and low birth weight irrespective of gestational age [11]. Some studies observed an association between maternal fever and other congenital anomalies (e.g., congenital heart defects and orofacial clefts) [12,13]. The best protection against influenza during pregnancy is vaccination. Immunization during pregnancy is a significant aspect of perinatal care. Influenza vaccination is vital in every influenza season to help reduce the impact of respiratory illness on the community, and the overburdened healthcare system, as the battle against the COVID-19 pandemic continues. The WHO has emphasized that pregnant women are the highest priority group for influenza vaccination and has thus recommended their vaccination. The CDC Advisory Committee on Immunization Practices recommends that all women who are or might be pregnant, or are in their postpartum period, during the influenza season should receive any licensed, age-appropriate, recommended inactivated influenza vaccine or the recombinant quadrivalent influenza vaccine, regardless of trimester [14].

Several studies conducted by the CDC and its partners support the safety of the flu vaccine for pregnant people and their babies. Some retrospective studies evaluating maternal safety found no correlation between influenza vaccines and maternal adverse events. More recently, three systematic studies published by the WHO did not detect an increased risk of miscarriage, fetal death, mortality, preterm birth, or congenital anomalies among pregnant women who received the flu vaccine [15,16,17]. Maternal vaccination can also protect a newborn from influenza after birth (due to maternal antibodies that pass to the developing fetus through the placenta during gestation). Breastfeeding women can also get a shot to protect themselves from the flu. Vaccination reduces parents’ risk of getting sick and passing the flu on to their babies, thus shielding their offspring from infection. This is particularly significant for children younger than 6 months old since they are too young to receive a flu vaccine themselves [18].

Despite the recommendations for vaccination against influenza during pregnancy, vaccination rates remain low due to concerns about the safety of the vaccine and fear of genetic abnormalities [19]. Unbiased maternal care providers (MCPs), obstetrician–gynecologists, and midwives should be distinctively located to increase maternal vaccination acceptance. According to studies, one of the reasons for the low vaccination rates among pregnant women is that they do not receive a clear recommendation from maternity care providers [20]. Other comparable studies have also noted that health professionals were more likely to recommend vaccination in pregnancy if they would personally have received the influenza vaccines of their own free will and/or if they as healthcare workers had the influenza vaccine shots [21,22,23].

Most countries strongly recommend that healthcare workers be vaccinated seasonally against influenza to protect themselves and their patients. Ongoing assessment of influenza vaccine effectiveness is critical to inform public health policy. Public health messaging should highlight the overall benefit of influenza vaccines so as to avoid an unprecedented disruption in healthcare systems around the globe. 

As the Coronavirus Disease 2019 (COVID-19) pandemic still causes life-threatening conditions, and the 2022–2023 influenza season epidemic comes to an early start in the European region, this study aims to record the knowledge and attitudes about influenza vaccination and explore healthcare professionals’ recommendations during the perinatal period. 

## 2. Materials and Methods

A cross-sectional survey was conducted from November 2020 to January 2021. A total of 240 health professionals attended the study from different scientific fields, e.g., midwives, obstetricians–gynecologists, and pediatricians, practicing in two of the largest public maternity hospitals in Athens, Greece.

Self-administered questionnaires were originally distributed either online or in a paper-and-pen format in person or through the mail. The study was initially designed with the questionnaires distributed in paper-and-pen format. The rapid increase in the COVID-19 pandemic crisis in Greece, with repeated lockdowns following research approval in September 2020, and the extremely limited number (only twenty) of hard copies collected so far, forced the authors to turn to scientific associations in order to amass participants through informative emails. Thus, the questionnaire was distributed by word-of-mouth to professionals working in the reference hospitals but also by sharing the research with the respective associations of each scientific field (the Hellenic Midwives’ Association and The Medical Association of Athens). The scientific associations informed their members via email of this research and shared the study questionnaire’s link with them. Consequently, two hundred and twenty questionnaires were administered exclusively online.

All questions were standardized so that all respondents received the same questions with identical wording. The COVID-19 pandemic has proven relentlessly challenging for healthcare workers and has strained the healthcare system in Greece in unprecedented ways. Limitations in staffing, high workload, heightened levels of somatic symptoms, and burnout led the researchers of this study to send the questionnaires by mail. This method was considered preferable due to time efficiency; further, respondents would not feel pressured and could answer when they had time, giving more accurate answers. 

The questionnaire was divided into three sections and was administered online and distributed via email. Section 1 included items on socio-demographics (age, gender) and practice characteristics (occupation/specialty, section of work, educational level). Section 2 included items on knowledge about the flu and modes of transmission (what is influenza, how it is caused, etc.). Section 3 included knowledge of the influenza vaccine (recommendations and guidelines about vaccination). The response rate was 80%. The questionnaires were sent to three hundred health professionals and, ultimately, two hundred and forty were returned fully completed. 

Participation in the survey was voluntary. A short paragraph was included at the beginning of the questionnaire to inform participants of the study’s objectives and their responses’ confidentiality. All participants gave informed consent. Data were collected anonymously and participants had the right to access their answers and withdraw from the research whenever they wished to. The study protocol was approved by the Research Ethics Boards of participating institutions (Clinical Research and Ethics Committee of ELENA VENIZELOU & ALEXANDRA Hospital, T59-Μ10/16-09-2020). The authors declare that the study procedures were followed according to the regulations established by the Clinical Research and Ethics Committee and the Helsinki Declaration of the World Medical Association.

Continuous variables are presented as mean (standard deviation). The normal distribution of variables was assessed using the Kolmogorov–Smirnov test due to the study’s sample size (˃70). The Kolmogorov–Smirnov test is a nonparametric goodness-of-fit test and is used to determine whether two distributions differ, or whether an underlying probability distribution differs from a hypothesized distribution.

All continuous variables were not normally distributed; therefore, their comparison was performed using an unpaired non-parametric two-tailed Mann–Whitney test on variables with two groups and the Kruskal–Wallis test on variables with three or more groups. Categorical variables were examined using Fisher’s exact or chi-square tests and are shown as absolute numbers (frequency percent); *p*-values under 0.05 were defined as significant. Statistical analysis was conducted using IBM SPSS-Statistics version 26.0 (IBM, Armonk, NY, USA).

## 3. Results

A total of 240 maternity care providers fully responded to the questionnaire. The mean age of the participants was 38.96 ± 9.56 years. As regards the occupation and specialty, 191 (79.6%) were midwives and 49 (20.4%) were medical doctors, of whom 27 (11.3%) were obstetricians–gynecologists and 22 (9.2%) were pediatricians. More than half (54.2%) of our population was working in public hospitals, 80 (33.3%) were working in private clinics and 30 (12.5) had their own private practices.

Regarding the educational level, 138 (57.5%) of the participants had bachelor’s degrees, 91 (37.9%) had master’s degrees, and 11 (4.6%) were Ph.D. holders. The mean amount of work experience was 11.45 ± 8.48 years (the characteristics of the study population are summarized in Table 1).

A total of 228 (95%) responders noted that the flu is a viral contagious infection affecting the respiratory system; 88 (36.7%) responders answered that it is caused by RNA viruses; and 63 (26.2%) replied that the types that cause the disease are A, B, and C. Most of the participants (221, 92.1%) mentioned that the flu is more serious than the common cold. Regarding the question “Which of the following is the official influenza crisis response plan?”, 17 (7.1%) respondents chose PERSEUS, 2 (0.8%) selected SOSTRATOS, 33 (13.8%) picked ARTEMIS, and 2 (0.8%) chose ATHENA. With reference to the questions about the transmission of the flu, the majority of participants gave the right answers (Data on the knowledge of health professionals about the flu and the ways of transmission is summarized in Table 2).

Numerous healthcare providers in our study (117, 73.8%) stated that they are aware of the guidelines of the Hellenic National Public Health Organization regarding flu vaccination, and more than half (150, 62.5%) mentioned that they are up to date on the developments surrounding the flu vaccine. The vast majority of our sample (233, 97.1%) declared that they are aware that the Hellenic National Public Health Organization recommends influenza vaccination for healthcare professionals; nonetheless, a great number (109, 45.4%) had not been vaccinated against the seasonal flu during the 2020 to 2021 season. Regarding the population groups in which vaccination is required, the most frequent answer was pregnant women (187 participants, 77.9%). Remarkably, 137 (57.1%) of our participants answered that vaccination is recommended throughout pregnancy, and 103 participants (42.9%) replied that vaccination is suggested either in the 1st or the 2nd, or in the 3rd trimester of pregnancy, or they did not know the correct answer.

The protection of their family appeared as the strongest motive for getting the vaccine (189 responders, 79.1%), followed by intending to protect their pregnant patients (173 responders, 73.3%), and themselves (172 responders, 72%). Concerns about side effects prevent a high percentage (118 respondents, 49.2%) from getting the vaccine, followed by the statement that the vaccine is not readily available (106 respondents, 44.2%). Almost all of the study’s population recommend the flu shot during pregnancy (227 respondents, 94.6%).

Furthermore, most participants were certain that the SARS-CoV-2 pandemic will increase influenza virus vaccine recommendations and vaccination rates (204 responders, 85%). Finally, the most recommended strategies regarding the promotion of flu vaccination were better informing health professionals about the vaccine and its recommendations (212 participants, 88.3%), vaccine information campaigns in the general population (120 participants, 50%), and an automatic vaccination reminder system (151 participants, 62.9%) (Data on health professionals’ knowledge of the influenza vaccine is summarized in Table 3).

### Association between Responders’ Characteristics and Knowledge about the Flu Vaccine

Most responders who mentioned that they were up to date on the developments surrounding the flu vaccine were obstetricians–gynecologists (*p* = 0.016); nevertheless, those who knew that the flu vaccine is recommended for the general population were midwives (*p* = 0.021) (Figure 1).

Pediatricians in our sample declared that vaccination is allowed for children older than six months (*p* = 0.01). Midwife participants answered that vaccination is recommended throughout the entire pregnancy (*p =* 0.02) (Figure 2).

Table 4 summarizes the associations between occupation/specialty and knowledge about the flu vaccine.

Remarkably, the respondents who stated that the vaccination is recommended through the entire pregnancy and those who had gotten the flu vaccine during the 2020–2021 season had their own private settings (*p* = 0.03 and *p* = 0.39, respectively).

Regarding work experience, the participants who recommended the flu vaccine had greater work experience (11.73 ± 8.54 years vs. 6.71 ± 5.79 years, *p* = 0.026) (Figure 3).

Table 5 summarizes the associations of the section of work and years of service with knowledge about the flu vaccine.

Moreover, a statistically significant number of our study population mentioned that they were aware of the guidelines of the Hellenic National Public Health Organization regarding flu vaccination, they were up to date on the developments surrounding the flu vaccine, knew that the Hellenic National Public Health Organization recommends influenza vaccination for healthcare professionals and the ones that recommend the flu vaccine were attending a statistically higher number of conferences/seminars/workshops per year (*p* = 0.013, *p* = 0.01, *p* = 0.02, and *p* = 0.013, respectively). Table 6 summarizes the relationships between the number of conferences/seminars/workshops attended per year and flu vaccine knowledge.

Finally, of great interest is the finding that there was a statistically significant difference in the proportion of participants who were vaccinated against the flu among the participants who recommended the vaccine (57.7% vaccinated vs. 42.3% unvaccinated, *p* = 0.01) (Figure 4).

Similarly, the participants who were aware of the guidelines of the Hellenic National Public Health Organization regarding flu vaccination and those who were up to date on the developments surrounding the flu vaccine remained more likely to be vaccinated against the seasonal flu during the 2020–2021 season (*p* = 0.002, and *p* = 0.011, respectively).

In addition, there was a statistically significant association between the number of healthcare professionals who recommended the vaccine and the number of healthcare professionals who had been informed about the flu vaccine (*p* = 0.01). Moreover, there was a statistically significant association between the number of healthcare professionals who recommended the vaccine and the number of healthcare professionals who were aware of the Hellenic National Public Health Organization guidelines regarding flu vaccination (*p* = 0.044). Table 7 summarizes the associations between information about the flu vaccine and vaccine recommendation/vaccination of healthcare professionals.

## 4. Discussion

Vaccination during pregnancy remains a national and international priority for maintaining perinatal health. Understanding healthcare providers’ knowledge about flu vaccination and attitudes toward vaccine acceptance is important in explaining current vaccination attainment levels among pregnant women. The purpose of this study was to identify reasons for the lack of vaccine uptake, which possibly includes a wide range of misconceptions or a lack of knowledge about influenza infection or the flu vaccine.

Due to the Hellenic National Public Health Organization in Greece, since 2015, there has been a gradual increase in staff vaccination coverage in both hospitals and Primary Health Care Centers [24]. In our sample, influenza vaccine coverage of healthcare providers was 54.6%, comparable to the percentage found by Vishram et al. in a UK study (58%) [25]. The majority of our sample (186, 77.5%) was unaware of the National Operational Plan to deal with an influenza pandemic, the so-called “ARTEMIS” plan. This was a thought-provoking finding because this plan is the same as the one for managing the COVID-19 pandemic in Greece. 

While obstetricians–gynecologists in our research stated that they were up to date on the developments surrounding the flu vaccine, those who correctly replied that the vaccination is recommended throughout the entire pregnancy were the participant midwives. A similar cross-sectional study conducted among midwives practicing in Paris showed that they were aware that vaccination against influenza is recommended during pregnancy (190/208, 91%) and can be administered during any trimester (155/208, 82%). Equally, in our study, 57.1% of the participants knew that flu vaccination is recommended throughout pregnancy, and the majority of those who gave the correct answer were midwives.

The health professionals participating in this study who recommended the flu vaccine during pregnancy had many years of work experience in maternity care. Similarly, in a UK survey, the more experienced a maternity care provider, the more confident they were in advice-giving regarding influenza vaccination [25].

In our study, it was found that the majority of healthcare professionals who recommend the vaccine had been informed about the flu vaccine, and they were also aware of the guidelines of the Hellenic National Public Health Organization regarding flu vaccination. This was a similar finding to a study at a regional hospital in the northeast of the Republic of Ireland where a correlation between healthcare professionals’ awareness of guidelines and flu vaccine recommendations was proved [22]. A cross-sectional survey among French midwives practicing in public and private sectors also demonstrated that a higher level of knowledge and the existence of a vaccination protocol against influenza were associated with higher offer and prescription rates [21].

Our responders who recommend the flu vaccine attended a statistically higher number of conferences/seminars/workshops annually. Ongoing professional education is vital to update healthcare providers’ knowledge in crucial scientific areas of interest so as to improve their ability to serve their clients.

Despite the fact that most of our sample declared that they were aware of the Hellenic National Public Health Organization influenza vaccination recommendation for healthcare professionals (233, 97.1%), a great proportion (109, 45.4%) had not been vaccinated against the seasonal flu during the 2020–2021 season. Furthermore, it is worth mentioning that in our study, 73.8% of our participants were fully cognizant of the guidelines of the Hellenic National Public Health Organization regarding flu vaccination and 62.5% were up to date on the developments surrounding the flu vaccine. This finding was analogous to another study in which the majority (95%) of healthcare professionals were aware of the health service executive guidelines on immunization; nonetheless, more than 75% of them did not receive the influenza vaccinations themselves and had no plans to receive it [22].

Worries about side effects were most commonly cited as the reason for not being vaccinated (118, 49.2%). This fact possibly explains the low rate of vaccination among our participants. This was also found in another study in England where 43% of healthcare professionals answered that concerns about side effects or personal illness/allergy were the reasons for not accepting influenza vaccination [25]. In a cross-sectional survey among midwives in Paris, the most frequent reason for non-vaccination was “not being worried about catching influenza” (33%) [21]. Studies consistently advise that when recommendations for influenza vaccination during pregnancy originate directly from a woman’s obstetrician–gynecologist or midwife and if the vaccine is available in the health professional’s office, the odds of vaccine acceptance and receipt are 5-fold to 50-fold higher [26,27]. Thus, it is critically important that all maternity care providers recommend and advocate for the influenza vaccine.

As in former studies, the present study had its own limitations. First, this research had a cross-sectional design meaning the study was conducted in a specific time period. Therefore, the timing of the snapshot is not guaranteed to be representative and the study cannot be used to analyze behavior over a period of time.

Another limitation of our research is the relatively small sample size which may affect the generalizability of our results. This study was carried out amid the COVID-19 pandemic and consequently, numerous factors, such as irregular and long working hours, shift-working system, role ambiguity, role conflict, and cancellation of their annual leave, operated as deterrents for the more massive participation of health workers. However, those who eventually participated in the survey completed the entire questionnaire, with no missing data. This study was not very flexible as researchers were generally confined to a single instrument for collecting data. Depth was also a problem with this survey. Research questions were standardized; thus, it often seemed difficult to ask anything other than general questions that a broad range of health professionals would understand.

Likewise, another limitation is convenience sampling, which proved efficient and simple to implement under the pandemic circumstances; however, the sample lacks clear generalizability. The last limitation was the unbalanced number of maternity care professionals. The vast majority of participants in our research were midwives (191 (79.6%)). This may have caused the study results to be more influenced by midwives’ views and to reflect mainly their knowledge and attitudes. Studies, where the professional distribution is equal or close, are required in the future.

Although the findings should be interpreted with caution, this study has several strengths. The results contribute to our understanding of three different groups of maternity care professionals (obstetricians–gynecologists, pediatricians, and midwives) who have been found to experience difficulty in dealing with many issues during the COVID-19 pandemic. More specifically, our research is the only one that studied these three specialties compared to other studies during the COVID-19 pandemic. As there are limited data on influenza vaccination uptake and determinants of uptake in obstetric populations, our research embraces the gap of knowledge and attitudes of primary maternity care providers on that issue, as influenza during pregnancy can be potentially life-threatening.

Even though our survey was conducted during the COVID-19 pandemic, it is noteworthy that the attitudes and knowledge on influenza vaccination among healthcare professionals were similar to those found in other studies conducted before the COVID-19 pandemic.

Notwithstanding the relatively limited sample, this work offers valuable insights into how maternity care professionals that have been at the frontline in the fight against the COVID-19 crisis are actively involved in providing good quality maternity care while managing this crisis.

This study has provided a deeper insight into an increasing necessity for personal and professional interventions that can strengthen the personal empowerment of maternity health professionals concerning influenza vaccination. Evidence-based information will guarantee a well-informed health workforce, leading to quality maternity services. Information delivery methods about the benefits of vaccination during pregnancy should be improved to increase the likelihood that someone will act on it [28]. Future health promotion campaigns may use the information in this study to address the concerns in support of influenza vaccination so as to improve vaccination uptake, following the new influenza A virus outbreak globally.

## 5. Conclusions

This study outlined reasons for the lack of vaccine uptake among health professionals working in maternity care, which mostly included a wide range of misconceptions or lack of knowledge about influenza infection or convenient access to vaccinations. There was a statistically significant difference in the percentage of participants from different scientific areas who responded that vaccination is recommended throughout the entire pregnancy, with midwives exhibiting the highest percentage. Given that the central role of Ob/Gs and midwives in the context of primary maternity care is prevention with the intention of improving maternal health and promoting safe motherhood, misconceptions about influenza and influenza vaccines should be improved by better educating healthcare workers. Health professional associations should ease the process of delivering information on perinatal vaccination to the end user in a way that is most likely to be understood and most likely to be acted upon. Strategies such as vaccine information campaigns in the general population and better-informing health professionals about the vaccine and its recommendations would help in the promotion of flu vaccination, as the European Commission has already published a communication on preparing for the autumn and winter 2022–2023 to help countries prepare their response to an expected increase of COVID-19 and influenza.

## Figures and Tables

**Figure 1 vaccines-11-00168-f001:**
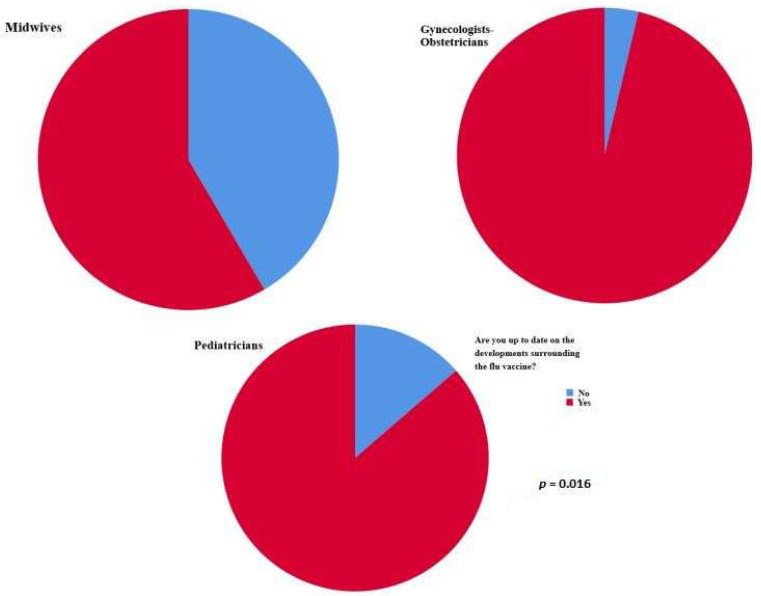
Most responders who mentioned that they were up to date on the developments surrounding the flu vaccine were obstetricians–gynecologists (*p =* 0.016).

**Figure 2 vaccines-11-00168-f002:**
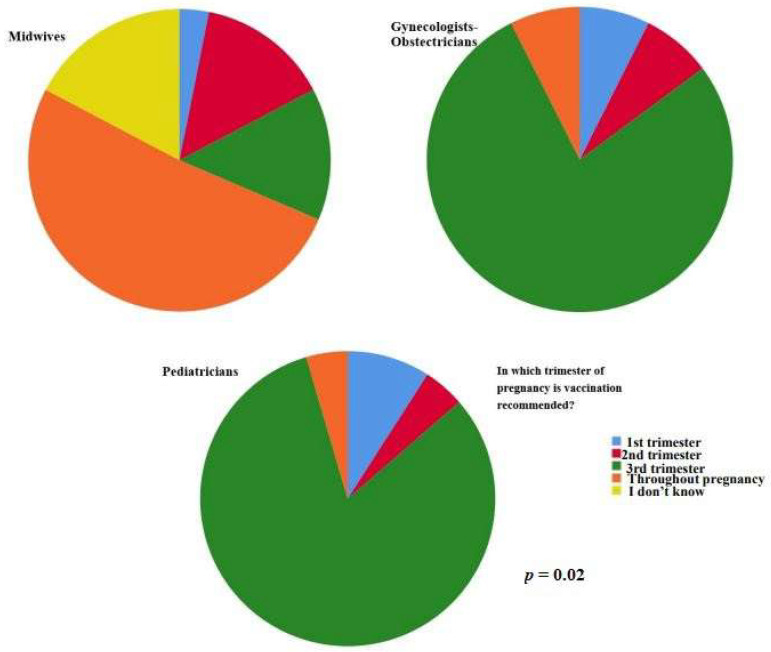
Midwife participants answered that vaccination is recommended throughout the entire pregnancy (*p =* 0.02).

**Figure 3 vaccines-11-00168-f003:**
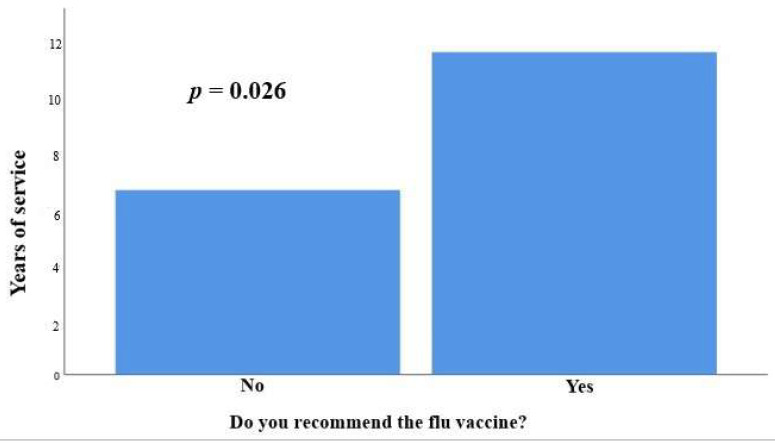
The participants who recommended the flu vaccine had greater work experience (11.73 ± 8.54 years vs. 6.71 ± 5.79 years, *p* = 0.026).

**Figure 4 vaccines-11-00168-f004:**
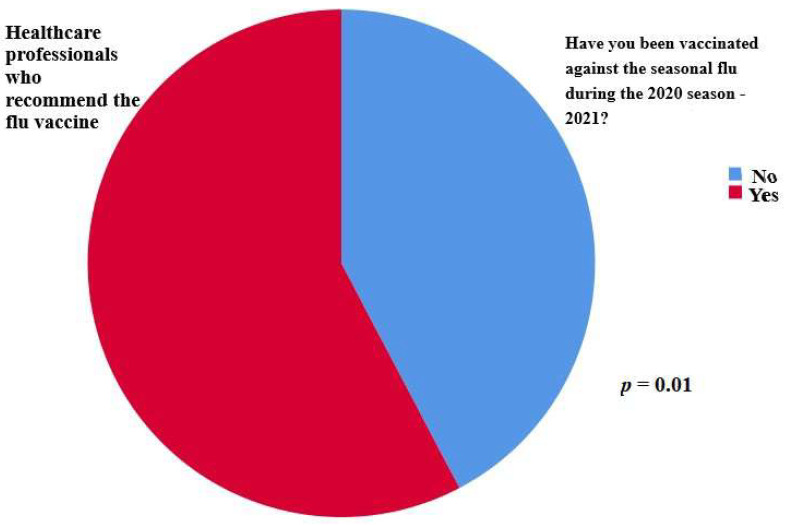
A statistically significant difference was observed in the proportion of participants who were vaccinated against the flu among the participants who recommended the vaccine (57.7% vaccinated vs. 42.3% unvaccinated, *p* = 0.01).

**Table 1 vaccines-11-00168-t001:** Demographics of the study population.

Variable	Mean	Standard Deviation
**Age (years)**	38.96	9.56
**Years of service**	8.48	11.45
**Number of conferences/seminars/workshops attended per year**	3.96	4.59
**Gender**	**N**	**%**
Female	212	88.3
Male	28	11.7
**Marital status**		
Single	99	41.2
Married	132	55
Other	9	3.8
**Number of children**		
0	109	45.4
1	45	18.8
2	71	29.6
3	12	5
4	3	1.3
**Age of older child**		
0–6 months	8	3.3
6 months–2 years	15	6.3
2 years–10 years	47	19.6
>10 years	54	22.5
**Profession/specialty**		
Midwife	191	79.6
Gynecologist–Obstetrician	27	11.3
Pediatrician	22	9.2
**Section of work**		
Public hospital	130	54.2
Private clinic	80	33.3
Private practice	30	12.5
**Educational level**		
Bachelor	138	57.5
Master	91	37.9
Doctorate	11	4.6
**Use of the Internet as a means of information and additional medical knowledge**		
No	29	12.1
Yes	211	87.9
**Attendance of conferences/seminars/workshops per year**		
No	23	9.6
Yes	217	90.4

**Table 2 vaccines-11-00168-t002:** Healthcare professionals’ knowledge about the flu and methods of transmission and prevention.

Question		
What is influenza, and how is it caused? (Multiple responses are allowed)	N	%
Viral contagious infection affecting the respiratory system	228	95
Caused by RNA viruses	88	36.7
The types of the virus are A, B, C, D	17	7.1
The types that cause the disease are A, B and C	62	25.8
I don’t know	4	1.7
**Is the flu more serious than a “common cold”?**		
Yes	221	92.1
No	15	6.3
I don’t know	4	1.7
**Have you received any information from competent bodies on issues related to the flu?**		
No	170	70.8
Yes	70	29.2
**Which of the following is the official crisis response plan for influenza?**		
PERSEUS	17	7.1
SOSTRATOS	2	0.8
ARTEMIS	33	13.8
ATHENA	2	0.8
I don’t know	186	77.5
**What are the ways of transmission of the flu?** **(Multiple responses are allowed)**		
Cough	235	97.9
Sneezing	232	96.7
Through the hands	202	84.2
Blood	16	6.7
Body fluids	41	17.1
**Do you think the chance of flu transmission is higher in the hospital?**		
Yes	156	65
No	75	31.3
I don’t know	9	3.8
**Do you think the flu is more likely to be spread in crowded places?**		
Yes	239	99.6
No	1	0.4
**Are healthcare professionals less vulnerable to influenza infections than other people?**		
Yes	29	12.1
No	207	86.3
I don’t know	4	1.7
**Can healthcare professionals spread the flu even when they feel well but are sick?**		
Yes	230	95.8
No	5	2.1
I don’t know	5	2.1
**Can people who are sick with the flu virus spread it only after they develop symptoms?**		
Yes	37	15.4
No	199	82.9
I don’t know	4	1.7
**For how many days can a person who has the flu transmit it?** **(Multiple responses are allowed)**		
One day before the onset up to 5–7 days after the onset of symptoms?	170	70.8
From the moment symptoms appear up to 5–7 days	61	25.4
Children and severely immunosuppressed patients may transmit for more than a week	66	27.5
I don’t know	13	5.4
**Do you know what are the necessary measures to take in order not to transmit the flu virus?** **(Multiple responses are allowed)**		
Use a tissue when coughing/sneezing	207	86.3
Regular hand washing using antiseptic	220	91.7
Washing hands after contact with patients	233	97.1
Washing with soap and hot water at the highest possible temperature, objects (utensils, sheets, towels, etc.) used by a patient with the flu in order to reuse them	155	64.6
Washing objects (dishes, sheets, towels, etc.) with antiseptic substances	113	47.1
Use of mask	220	91.7
Use of gloves	173	72.1
Use of protective glasses	0	0
Use of disposable blouse/apron	127	52.9
Disinfection of multi-use materials (sphygmomanometer, etc.)	171	71.3
Isolation of patients with influenza who require hospitalization, co-hospitalization of these patients (cohorting)	172	71.7
Avoiding unnecessary movement of patients in public places	195	81.3
In case of necessary transport of a patient, use of a mask also by him	189	78.8
Avoiding patient visits	0	0
I don’t know	0	0

**Table 3 vaccines-11-00168-t003:** Health professionals’ knowledge of the influenza vaccine.

Question		
**Do you know there is a vaccine for the flu?**	**N**	**%**
Yes	240	100
No	0	0
**Have you been informed about the flu vaccine?**		
Yes	220	91.7
No	20	7.6
**If so, how did you learn about the flu vaccine? (Multiple responses are allowed)**		
Hellenic National Public Health Organization instructions	61	25.4
Hospital training program	74	30.8
Leaflet or posters	35	14.6
Seminars—Speeches	49	20.4
Media	44	18.3
**Does the flu vaccine contain live viruses that can cause people to get the flu?**		
Yes	58	24.2
No	160	66.7
I don’t know	22	9.2
**Are you aware of the guidelines of the Hellenic National Public Health Organization regarding flu vaccination** **?**		
Yes	177	73.8
No	63	25.9
**Are you up to date on the developments surrounding the flu vaccine?**		
Yes	150	62.5
No	89	37.1
**Does the Hellenic National Public Health Organization recommend influenza vaccination for healthcare professionals?**		
Yes	233	97.1
No	7	2.9
**In which population groups is vaccination necessary? (Multiple responses are allowed)**		
In people older than 50 years	154	64.2
In adults with a Body Mass Index (BMI) > 40 kg/m^2^	144	60
In children, who are over 6 months old and suffer from diabetes, chronic heart and lung diseases	169	70.4
In children taking long-term aspirin (e.g., Kawasaki disease, rheumatoid arthritis, etc.) to reduce the risk of developing Reye’s syndrome after the flu	134	55.8
In pregnant women	187	77.9
In lactating women	136	56.7
To those suffering from chronic diseases	214	89.2
In people who are immunosuppressed	192	80
To people who are in contact with high-risk people	207	86.3
I don’t know	9	3.8
**From what age is vaccination allowed for children?**		
After the 1st year of life	94	39.2
After 6 months of life	146	60.8
**In which trimester of pregnancy is vaccination recommended?**		
1st trimester	10	4.2
2nd trimester	29	12.1
3rd trimester	28	11.7
Throughout pregnancy	137	57.1
I don’t know	36	15
**Have you been vaccinated against the seasonal flu during the 2020–2021 season?**		
Yes	131	54.6
No	107	45.4
**Reasons for vaccination against the influenza virus (Multiple responses are allowed)**		
The flu vaccine is effective	148	61.7
The flu shot is safe	159	66.3
The flu shot is free	102	42.5
Influenza is a serious illness	102	42.5
To protect myself	172	71.7
I suffer from a chronic illness	81	33.8
To protect my family	189	78.8
Encouragement from the workplace	96	40
Desire for immunization due to work	157	65.4
To protect my patients	173	72.1
**Reasons for not vaccinating against the influenza virus** **(Multiple responses are allowed)**		
It is not efficient	46	19.2
It is not safe	49	20.4
Worry about side effects	118	49.2
The vaccine causes influenza	23	9.6
The vaccine costs	24	10
Not readily available	106	44.2
Insufficient information about the vaccine	50	20.8
Prevention by others	52	21.7
I forget/don’t have time	66	27.5
It is not necessary for me	45	18.8
Flu is not a serious illness	23	9.6
I have a phobia of needles	11	4.6
I’m worried I’m going to hurt	39	16.3
I am against vaccines in general	52	21.7
I belong to a population group that is not allowed to be vaccinated	86	35.8
Allergy to previous vaccination	18	7.5
Pregnancy	18	7.5
Breastfeeding	1	0.4
**Do you think there is sufficient information for health professionals about the use of the flu vaccine?**		
Yes	91	37.9
No	141	58.8
I don’t know	8	3.3
**Do you think the SARS-CoV-2 pandemic will increase influenza virus recommendation and vaccination rates?**		
Yes	204	84
No	27	11.3
I don’t know	9	0.8
**Which of the following recommendations do you think would help promote flu vaccination? (Multiple responses are allowed)**		
Better inform healthcare professionals about the vaccine and its recommendations	212	88.3
Less workload for healthcare professionals	54	22.5
Greater and easier availability of vaccines	101	42.1
Automatic vaccination reminder system	151	62.9
To be imposed by the state	37	15.4
Vaccine information campaigns in the general population	120	50
I do not agree with the promotion of vaccination	5	2.1
**Do you recommend the flu vaccine?**		
Yes	227	94.6
No	13	5.4

**Table 4 vaccines-11-00168-t004:** Associations between occupation/specialty and knowledge about the flu vaccine.

Occupation/Specialty	Are You Aware of the Guidelines of the Hellenic National Public Health Organization Regarding Flu Vaccination?		Total	*p*
	No	Yes		
**Midwife**	54	136	190	0.211
**Obstetrician–Gynecologist**	5	22	27	
**Pediatrician**	3	19	22	
	**Are you up to date on the developments surrounding the flu vaccine?**		**Total**	** *p* **
	No	Yes		
**Midwife**	79	111	190	**0.016**
**Obstetrician–Gynecologist**	7	20	27	
**Pediatrician**	3	19	22	
	**Does the Hellenic National Public Health Organization recommend influenza vaccination for healthcare professionals?**		**Total**	*p*
	Yes	No		
**Midwife**	184	7	191	0.397
**Obstetrician–Gynecologist**	27	0	27	
**Pediatrician**	22	0	22	
	**From what age is vaccination allowed for children?**		**Total**	** *p* **
	After the first year of life	After the first six months of life		
**Midwife**	80	111	191	**0.010**
**Obstetrician–Gynecologist**	12	15	27	
**Pediatrician**	2	20	22	
	**In which trimester of pregnancy is vaccination recommended?**		**Total**	** *p* **
	Throughout entire pregnancy	Other		
**Midwife**	98	93	191	**0.020**
**Obstetrician–Gynecologist**	21	6	27	
**Pediatrician**	18	4	22	
	**Have you been vaccinated against the seasonal flu during the 2020–2021 season?**		**Total**	*p*
	No	Yes		
**Midwife**	94	97	191	0.065
**Obstetrician–Gynecologist**	8	19	27	
**Pediatrician**	7	15	22	
	**Do you recommend the flu vaccine?**		**Total**	*p*
	No	Yes		
**Midwife**	13	178	191	0.172
**Obstetrician–Gynecologist**	0	27	27	
**Pediatrician**	0	22	22	
	**Is the flu vaccine recommended for the general population?**			
	No	Yes	I don’t know	** *p* **
**Midwife**	46	140	5	**0.021**
**Obstetrician–Gynecologist**	11	15	1	
**Pediatrician**	12	10	0	
	**May the flu shot not be effective if the vaccine contains other types of the virus than the ones that are in an outbreak?**			
	No	Yes	I don’t know	*p*
**Midwife**	24	120	47	0.126
**Obstetrician–Gynecologist**	2	21	4	
**Pediatrician**	2	19	1	

*p*-value for the difference between groups was assessed using the Fisher’s exact test or chi-square test as appropriate.

**Table 5 vaccines-11-00168-t005:** Associations of sections of work and years of service with knowledge about the flu vaccine.

Section of Work	Are You Aware of the Guidelines of the Hellenic National Public Health Organization Regarding Flu Vaccination?		Total	*p*
	No	Yes		
**Public hospital**	40	90	130	**0.030**
**Private clinic**	21	59	80	
**Private practice**	2	28	30	
	**Are you up to date on the developments surrounding the flu vaccine?**		**Total**	*p*
	No	Yes		
**Public hospital**	54	76	130	0.298
**Private clinic**	28	52	80	
**Private practice**	8	22	30	
	**Does the Hellenic National Public Health Organization recommend influenza vaccination for healthcare professionals?**		**Total**	*p*
	Yes	No		
**Public hospital**	126	4	130	0.961
**Private clinic**	78	2	80	
**Private practice**	29	1	30	
	**From what age is vaccination allowed for children?**		**Total**	*p*
	After the first year of life	After the first six months of life		
**Public hospital**	52	78	130	0.941
**Private clinic**	31	49	80	
**Private practice**	11	19	30	
	**In which trimester of pregnancy is vaccination recommended?**		**Total**	** *p* **
	Throughout entire pregnancy	Other		
**Public hospital**	75	55	130	**0.030**
**Private clinic**	39	41	80	
**Private practice**	23	7	30	
	**Have you been vaccinated against the seasonal flu during the 2020–2021 season?**		**Total**	** *p* **
	No	Yes		
**Public hospital**	67	63	130	**0.039**
**Private clinic**	34	46	80	
**Private practice**	8	22	30	
	**Do you recommend the flu vaccine?**		**Total**	*p*
	No	Yes		
**Public hospital**	7	123	130	0.834
**Private clinic**	5	75	80	
**Private practice**	1	29	30	
	**Is the flu vaccine recommended for the general population?**			
	No	Yes	I don’t know	*p*
**Public hospital**	34	93	3	0.224
**Private clinic**	21	57	2	
**Private practice**	14	15	1	
	**May the flu shot not be effective if the vaccine contains other types of the virus than the ones that are in an outbreak?**			
	No	Yes	I don’t know	*p*
**Public hospital**	16	85	29	0.484
**Private clinic**	11	51	18	
**Private practice**	1	24	5	
**Time of service (years) ± Standard Deviation**	**Are you aware of the guidelines of the Hellenic National Public Health Organization regarding flu vaccination?**			*p*
	No 10.39 ± 8.5	Yes 11.70 ± 8.39		0.186
	**Are you up to date on the developments surrounding the flu vaccine?**			*p*
	No	Yes		0.077
	10.23 ± 8.12	12.05 ± 8.54		
	**Does the Hellenic National Public Health Organization recommend influenza vaccination for healthcare professionals?**			*p*
	Yes	No		0.864
	11.42 ± 8.47	12.28 ± 9.42		
	**From what age is vaccination allowed for children?**			*p*
	After the first year of life	After the first six months of life		0.089
	12.56 ± 8.67	10.71 ± 8.31		
	**In which trimester of pregnancy is vaccination recommended?**			*p*
	Throughout entire pregnancy	Other		0.906
	11.23 ± 7.92	11.73 ± 9.22		
	**Have you been vaccinated against the seasonal flu during the 2020–2021 season?**			*p*
	No	Yes		0.125
	10.81 ± 8.96	11.97 ± 8.06		
	**Do you recommend the flu vaccine?**			** *p* **
	No	Yes		**0.026**
	6.71 ± 5.79	11.73 ± 8.54		
	**Is the flu vaccine recommended for the general population?**			*p*
	No	Yes	I don’t know	0.112
	10.82 ± 7.68	11.92 ± 8.8	6.16 ± 7.57	
	**May the flu shot not be effective if the vaccine contains other types of the virus than the ones that are in an outbreak?**			*p*
	No	Yes	I don’t know	0.150
	8.48 ± 6.62	12.04 ± 8.75	11.22 ± 8.48	

*p* value for the difference between groups was assessed using the Fisher’s exact test or the chi-square test for categorical variables, Mann–Whitney test on continuous variables with two groups, and the Kruskal–Wallis test on continuous variables with three or more groups as appropriate.

**Table 6 vaccines-11-00168-t006:** Associations between the number of attended conferences/seminars/workshops per year and the knowledge about the flu vaccine.

	Are You Aware of the Guidelines of the Hellenic National Public Health Organization Regarding Flu Vaccination?			*p*
**Number of attended** **conferences/seminars/** **workshops per year ±** **Standard Deviation**	No 3.84 ± 6.97	Yes 4.02 ± 3.43		**0.013**
	**Are you up to date on the developments surrounding the flu vaccine?**			** *p* **
	No	Yes		**0.001**
	3.06 ± 3.39	4.5 ± 5.12		
	**Does the Hellenic National Public Health Organization recommend influenza vaccination for healthcare professionals?**			** *p* **
	Yes	No		**0.002**
	4.05 ± 4.63	1 ± 0.81		
	**From what age is vaccination allowed for children?**			*p*
	After the first year of life	After the first six months of life		0.108
	3.73 ± 5.55	4.1 ± 3.87		
	**In which trimester of pregnancy is vaccination recommended?**			*p*
	Throughout entire pregnancy	Other		0.121
	3.96 ± 3.41	3.95 ± 5.82		
	**Have you been vaccinated against the seasonal flu during the 2020–2021 season?**			*p*
	No	Yes		0.547
	4.1 ± 5.72	3.84 ± 3.4		
	**Do you recommend the flu vaccine?**			** *p* **
	No	Yes		**0.013**
	1.77 ± 1.30	4.08 ± 4.68		
	**Is the flu vaccine recommended for the general population?**			*p*
	No	Yes	I don’t know	0.555
	3.87 ± 3.06	4.02 ± 5.15	3.17 ± 2.85	
**May the flu shot not be effective if the vaccine contains other types of the virus than the ones that are in an outbreak?**	No	Yes	I don’t know	*p*
**Number of attended** **conferences/seminars/** **workshops per year ±** **Standard Deviation**	5.07 ± 4.30	4.10 ± 4.99	2.92 ± 3.09	0.150

*p*-value was assessed using the Mann–Whitney test on continuous variables with two groups and the Kruskal–Wallis test on continuous variables with three or more groups.

**Table 7 vaccines-11-00168-t007:** Associations between information about the flu vaccine and vaccine recommendation/vaccination of the healthcare professionals.

	Have You Been Informed about the Flu Vaccine?			
**Do you recommend the flu vaccine?**	No	Yes	Total	** *p* **
No	5	8	13	**0.01**
Yes	15	212	227	
	**Are you aware of the guidelines of the Hellenic National Public Health Organization regarding flu vaccination?**			
**Do you recommend the flu vaccine?**	No	Yes	Total	** *p* **
No	7	6	13	**0.044**
Yes	56	171	227	
	**Are you up to date on the developments surrounding the flu vaccine?**			
**Do you recommend the flu vaccine?**	No	Yes	Total	*p*
No	8	5	13	0.079
Yes	82	145	227	
	**Have you been vaccinated against the seasonal flu during the 2020–2021 season?**			
**Do you recommend the flu vaccine?**	No	Yes	Total	** *p* **
No	13	0	13	**0.01**
Yes	96	131	227	
	**Have you been informed about the flu vaccine?**			
**Have you been vaccinated against the seasonal flu during the 2020–2021 season?**	No	Yes	Total	** *p* **
No	15	94	109	**0.001**
Yes	5	126	131	
	**Are you aware of the guidelines of the Hellenic National Public Health Organization regarding flu vaccination?**			
**Have you been vaccinated against the seasonal flu during the 2020–2021 season?**	No	Yes	Total	** *p* **
No	40	69	109	**0.002**
Yes	23	108	131	
	**Are you up to date on the developments surrounding the flu vaccine?**			
**Have you been vaccinated against the seasonal flu during the 2020–2021 season?**	No	Yes	Total	** *p* **
No	51	58	109	**0.011**
Yes	39	92	131	

*p* value for the difference between groups was assessed using the Fisher’s exact test or the chi-square test as appropriate.

## Data Availability

The datasets generated during and/or analyzed during the current study are not publicly available; however, they are available from the corresponding author upon reasonable request.

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
