# Peer review of "Knowledge and Attitudes of Healthcare Professionals Regarding Perinatal Influenza Vaccination during the COVID-19 Pandemic"

_vaccines, 2023, doi:10.3390/vaccines11010168_

Round 1
Reviewer 1 Report
Known in the field based on previous literatures:
1. Influenza is an infection of the respiratory system mainly include nose, throat, and lungs and commonly called flu. Covid-19, an ongoing pandemic, is an infectious disease caused by severe acute respiratory syndrome coronavirus 2 (SARS-CoV-2) and some of the symptoms are like flu.
2. The flu vaccine comes in inactive and weakened viral forms and depending on the type they can be injected into a muscle, sprayed into the nose, or injected into the middle layer of the skin.
In this article authors mentioned following findings:
I have gone through the article titled “Knowledge and attitudes of health care professionals regarding perinatal Influenza vaccination during the Covid 19 pandemic”. Authors mentioned many information and the current developments regarding influenza vaccination and its prospects in Greece. The core points mentioned by authors are-
1. Authors identified the reasons for the lack of vaccine uptake including a wide range of misconceptions.
2. Authors also pointed out how to improve influenza vaccination.
The similar studies are already existed except numerical value, but the facts and material presented are interesting and generally supportive of the conclusions drawn. There are, however, some issues that require the authors' attention. The following suggestions if incorporated could help in the better understanding of the significance of the work and implications.
Minor/major Concerns:
1. I could not find the new things; hence, authors should clearly mention about how this article different from rest, except numerical value? Does it embrace a specific gap in the field as compared to previous data?
2. There are only tables hence authors can add bar plot of significant data.
3. Authors can reframe this sentence “The majority of participants in this study (186, 77.5%) did not know what the official crisis response plan for influenza is”.
Author Response
- I could not find the new things; hence, authors should clearly mention about how this article different from rest, except numerical value? Does it embrace a specific gap in the field as compared to previous data?
Thank you for your comment. We clearly stated in our research’s strengths, as you suggested, that our study is the first to combine the 3 major perinatal care providers (Ob/Gs, midwives and pediatricians) in order to investigate their knowledge and attitudes on influenza vaccination throughout pregnancy during the Covid-19 pandemic. As there are limited data on influenza vaccination uptake and determinants of uptake in obstetric populations our research embraces the gap of knowledge and attitudes of primary maternity care providers on that issue.
- There are only tables hence authors can add bar plot of significant data.
Thank you for your comment. We followed your suggestion and have presented some of our research findings as pie charts for a better understanding by potential readers of the manuscript. We consider it easier for readers to spot percentages in a pie chart than in a stacked bar or column chart.
- Authors can reframe this sentence “The majority of participants in this study (186, 77.5%) did not know what the official crisis response plan for influenza is”.
Thank you for your comment! The sentence was rephrased as recommended
Reviewer 2 Report
Please check the attached PDF file.

Author Response
Thank you for your valuable comments!
Please find attached the revisions requested.

Round 2
Reviewer 2 Report
Please check the attached PDF file.

Author Response
Respected Reviewers,
We would like to express our thanks to you for the time spent on our manuscript’s revision.
Your comments are highly appreciated and we resubmit the revised version according to your remarks. We have made some additions/corrections to the main text, and we listed them with line numbers in this rebuttal letter. We also highlighted all the revised sentences in yellow marker in the submitted manuscript so as to correspond to your comments and be easily readable by you.
Kind regards,
Major Comment
#1. Methods How did the authors recruit study participants?
#2. Questionnaire, paper version or online version?
#1 & #2. Thank you for your comments. We made some additions based on your remarks as follows:
“The study was initially designed to distribute the questionnaires in paper and pen format. The rapid increase of the COVID-19 pandemic crisis in Greece with repeated lockdowns after the research approval in September 2020, and the extremely limited number of hard copies collected so far, forced the authors to turn to scientific associations in order to amass participants through informative emails. Thus, the questionnaire was distributed by word of mouth to professionals working in the reference hospitals but also by sharing the research with the respective associations of each scientific field (the Hellenic Midwives’ Association & The Medical Association of Athens). The scientific associations informed their members via email of the research and shared the study questionnaire’s link with them. Consequently, the vast majority of the questionnaires were administered exclusively online.”
(Lines 118-128)
#3. #3. Comment on the analysis method.
Thank you for your comment. Following your advice, we highlighted the paragraph and made our additions rewriting it accordingly.
“The assessment of the normal distribution of variables was performed with the use of the Kolmogorov-Smirnov test as the study sample was large. The Kolmogorov–Smirnov test is a nonparametric goodness-of-fit test and is used to determine whether two distributions differ, or whether an underlying probability distribution differs from a hypothesized distribution.
All continuous variables were not normally distributed; therefore, their comparison was performed using an unpaired non-parametric two-tailed Mann-Whitney test on variables with two groups and the Kruskal–Wallis test on variables with three or more groups” (Lines 154-162)
The Kolmogorov–Smirnov test is a nonparametric goodness-of-fit test and is used to determine whether two distributions differ, or whether an underlying probability distribution differs from a hypothesized distribution. (Lines 156-158)
And we also highlighted that:
All continuous variables were not normally distributed (Line 159)
Please let us inform you that we chose the Kolmogorov–Smirnov test as we had samples coming from different populations (midwives, Ob/Gs, pediatricians) and the final number of our participants was ˃70.
Minor comment
#4. Tables
Corrections are made according to your instruction in the alignment of table 5 and in the lines 321-322 that you considered too long. Please check the sentence as follows:
“P value was assessed using the Mann-Whitney test on continuous variables with two groups and the Kruskal–Wallis test on continuous variables with three or more groups”. (Lines 323-324)
Round 3
Reviewer 2 Report
Please check the attached PDF file.

Author Response
Respected reviewer,
Thank you for your remarks. Please check our revisions according to your comments and we hope they will satisfy you.
Kind regards
#1. Questionnaire, paper version or online version?
Thank you for your comment. We have provided the required numbers of questionnaires in paper and online versions so as to be exact (Lines 121 & 127-128)
#2. Table 6
Thank you for your remark. We rewrote the line in table 6 as requested.